# Specialized Pro-Resolving Lipid Mediators in Neonatal Cardiovascular Physiology and Diseases

**DOI:** 10.3390/antiox10060933

**Published:** 2021-06-08

**Authors:** Andrea Gila-Diaz, Gloria Herranz Carrillo, Pratibha Singh, David Ramiro-Cortijo

**Affiliations:** 1Department of Physiology, Faculty of Medicine, Universidad Autónoma de Madrid, C/Arzobispo Morcillo 2, 28029 Madrid, Spain; andrea.gila@uam.es; 2Division of Neonatology, Hospital Clínico San Carlos, Instituto de Investigación Sanitaria del Hospital Clínico San Carlos (IdISSC), C/Profesor Martin Lagos s/n, 28040 Madrid, Spain; gherranz@gmail.com; 3Division of Gastroenterology, Beth Israel Deaconess Medical Center, Harvard Medical School, 330 Brookline Avenue, Boston, MA 02215, USA

**Keywords:** oxylipins, specialized pro-resolving mediators, long chain polyunsaturated fatty acids, newborn, premature infant, cardiovascular diseases

## Abstract

Cardiovascular disease remains a leading cause of mortality worldwide. Unresolved inflammation plays a critical role in cardiovascular diseases development. Specialized Pro-Resolving Mediators (SPMs), derived from long chain polyunsaturated fatty acids (LCPUFAs), enhances the host defense, by resolving the inflammation and tissue repair. In addition, SPMs also have anti-inflammatory properties. These physiological effects depend on the availability of LCPUFAs precursors and cellular metabolic balance. Most of the studies have focused on the impact of SPMs in adult cardiovascular health and diseases. In this review, we discuss LCPUFAs metabolism, SPMs, and their potential effect on cardiovascular health and diseases primarily focusing in neonates. A better understanding of the role of these SPMs in cardiovascular health and diseases in neonates could lead to the development of novel therapeutic approaches in cardiovascular dysfunction.

## 1. Introduction

Despite encouraging advancement in the field of public health, cardiology, and scientific discovery for the prevention and treatment, cardiovascular diseases are a leading cause of mortality and are one of the major public health concerns globally. Cardiovascular diseases are responsible for 30% of deaths worldwide [1,2,3]. Even after surviving myocardial infarction or stroke, the likelihood of developing secondary complications is high, and have significant health and economic burden through hospitalizations and follow-up clinical care.

In addition to lifestyle, nutrition also plays an important role in the etiology and treatment of cardiovascular diseases. Nutrient-based approaches have been suggested to reduce the risk of developing cardiovascular diseases by reducing various risk factors such as hyperlipidemia, diabetes, metabolic syndrome, overweight/obesity, and inflammatory mediators [4,5]. The cardioprotective effect of diet could be associated with an improved lipid profile, decreased blood pressure, antioxidant properties, and decreased platelet activation. It is now widely accepted that chronic inflammation plays a crucial role in the development of cardiovascular diseases. During infection and injury, innate immune cells are recruited to the site of tissue damage and produce classical eicosanoids that are highly pro-inflammatory. However, excessive pro-inflammatory response can cause cell damage and occasionally apoptosis/necrosis. Failure of macrophages to clear apoptotic cells prolongs the inflammation [6]. Resolution of inflammation is regulated by a class of bioactive lipid mediators, called specialized pro-resolving mediators. Impaired resolution of inflammation results in chronic inflammation that is associated with cancers, autoimmune diseases, metabolic, cardiovascular, and neurodegenerative diseases resulting in organ dysfunction [6,7]. Acute and chronic inflammation coexist over long periods. Uncontrolled inflammation and failure to resolve the inflammation is the underpinning of several prevalent human diseases, including cardiovascular diseases. Furthermore, finding new ways to target the inflammatory response is gaining attraction as a new therapeutic approach to treat these diseases [8].

Lipids are an important structural component of the cell membrane and the susceptibility of lipid membranes to oxidative damage is dependent on the degree of unsaturation of its fatty acids. Evidence suggests that dietary fatty acids, more specifically long-chain polyunsaturated fatty acids (LCPUFAs), are protective against cardiovascular diseases [9] and these beneficial effects are mediated by LCPUFAs derived lipid mediators [10,11]. LCPUFAs serve as precursors for eicosanoid production (prostaglandins, prostacyclins, thromboxanes, and leukotrienes), which are important regulators of thrombocyte aggregation, inflammatory response, leukocyte functions, vasoconstriction, vasodilatation, blood pressure, and bronchial constriction.

This review is aimed at discussing LCPUFAs metabolism, bioactive lipid mediators derived from LCPUFAs, and their therapeutic potential to the health benefits in cardiovascular health and diseases, particularly in the neonatal period, based on nutritional intervention health benefits of dietary LCPUFAs.

## 2. The Role of Fatty Acids in Cardiovascular Development and Physiology

The perinatal life is a critical period in the development and physiology of cardiac tissue. Higher cardiac energy metabolism is required in response to early postnatal oxygen levels, and increased cell proliferation is needed to attain normal neonatal growth and development [12,13,14]. Cardiomyocyte regeneration is a highly energy-consuming process, and changes in energy metabolism happen as the neonatal cardiac development evolves rapidly within a short postnatal period; afterward, the cardiomyocytes exit the cell cycle. Fetal cardiomyocytes use glycolysis as the main source of energy during proliferation [15,16]. During early postnatal development, there is a shift from glycolysis to using fatty acid oxidation (FAO) [17,18], possibly due to increased energy demand during the transitioning from a fetal to postnatal period [19], suggesting a correlation between cardiomyocyte proliferation and high oxidative energy metabolism. It is not clear whether this heart metabolic shift accounts for cardiomyocyte proliferation and hypertrophy or it is a consequence of increased oxygen availability in postnatal environment. Animal studies have shown that in younger animals, FAO facilitates cardiomyocyte proliferation and hypertrophic growth [16]. Altered FAO could be associated with cardiac diseases [20,21]. Increased levels of oxygen during transition, from fetal to postnatal period, results in significantly increased oxidative stress as a result of the production of reactive oxygen species (ROS) with subsequent damage to cardiac tissue. These data could suggest that reducing the oxygen level and increased antioxidant levels might be useful to maintain the proliferative capacity of neonatal cardiomyocytes for a longer time [12,22]. These data suggest that the antioxidant could be used as a relevant therapy to extend the critical time window for cardiomyocyte proliferation by upregulating mitochondrial biogenesis and decrease mitophagy [12,23].

Cardiolipin (CL) is a phospholipid that is exclusively present in the inner mitochondrial membrane, where it is essential for optimal functions of various key enzymes involved in mitochondrial respiration [24,25]. Linoleic acid is one of the major fatty acids of CL in human myocardium [26]. Evidence has shown that CL must contain four linoleic acid side chains for optimal mitochondrial function, and loss of linoleic acid content in CL resulted in human and rat models of heart failure [26]. Furthermore, diet supplementation containing high linoleic acid attenuated contractile failure by improving mitochondrial dysfunction in rat models of hypertensive heart failure [27].

### 2.1. LCPUFAs and Cardiovascular Diseases

LCPUFAs are a group of essential fatty acids that can classify into two main categories: omega-3 (n-3) and omega-6 (n-6), depending on the position of the first double bond from the methyl end group of the fatty acid. Linoleic acid (LA; 18:2n-6) and α-linolenic acid (ALA; 18:3n-3) are the precursors of n-3 and n-6 LCPUFAs, respectively. The metabolic pathway involves a series of desaturase and elongase enzymes leading to the production of eicosapentaenoic acid (EPA; 22:5n-3) from ALA or arachidonic acid (ARA; 20:4n-6) from LA, based on the availability. EPA is then elongated and desaturated to produce docosahexaenoic acid (DHA; 22:6n-3).

Human studies have shown limited endogenous synthesis of DHA from n-3 fatty acid precursor ALA, necessitating the exogenous supplementation of a diet containing n-3 LCPUFAs for optimal health and development [28,29]. In adults, circulating EPA levels increased with DHA supplementation. However, ALA and EPA supplementation did not result in increased DHA levels in circulation [29]. In contrast, Metherel et al. showed that supplemental EPA was converted to DHA [30]. Increased levels of DHA to EPA could be either due to retro-conversion [31,32,33] or reduced EPA metabolism [34], suggesting that circulating levels of EPA and DHA are differently regulated.

A large, prospective, randomized clinical trial of 11,324 patients with a recent myocardial infarction showed a reduced relative risk of death in response to LCPUFAs [35]. A lower rate of myocardial infarction was observed in people consuming an Inuit diet, a diet characterized by the high consumption of marine mammals and fish [36]. The Inuit diet is rich in n-3 LCPUFAs, particularly EPA and DHA compared to those consuming a Western diet, which is abundant in n-6 LCPUFAs, mainly derived from vegetable oils rich in LA [11,37]. Furthermore, a significant difference was observed for cardiovascular mortality in two cohorts with n-3 versus n-6 LCPUFA enriched diets [38,39]. Clinical studies have shown a reduction in mortality of patients with cardiovascular diseases in response to n-3 LCPUFAs [40,41]. Meta-analyses in adults have shown that the Mediterranean diet, enriched in n-3 LCPUFAs [42] and oleic acid [43], has a protective effect against cardiovascular diseases [43,44,45]. These studies declared the importance of bioactive compounds related to the diet. The beneficial effect of n-3 LCPUFAs (EPA and DHA) against cardiovascular diseases could be due to anti-arrhythmic, anti-inflammatory, anti-thrombotic, and hypolipidemic effects and improvement of vascular function [46,47]. However, a recent clinical trial failed to show a protective effect against arrhythmia in patients with myocardial infarction [48]. Supplementation of n-3 LCPUFAs showed no effect on atrial fibrillation [49], cardiac death, myocardial infarction, or stroke [50].

In the Mediterranean diet, one of the main dietary source of n-3 LCPUFAs is fish, particularly oily fish species and shellfish [51]. Neonates, specifically preterm infants, are at risk of a LCPUFA deficit, which has been associated with a subsequent increased risk of various neonatal diseases [52]. Infants with heart disease may require longer parenteral nutrition during their hospitalization that fails to maintain the critical blood levels of LCPUFAs. In neonates, adequate nutritional intake of LCPUFAs is important for visual development, growth, complex brain function, and immune function [53]. Parenteral administration of n-3 LCPUFAs containing lipid emulsion to infants previous to cardiac surgery decreased pro-inflammatory response after surgery [54]. These data suggest the beneficial impact of LCPUFAs in infants with cardiac diseases [52].

### 2.2. LCPUFAs Derived Mediators and Their Implication in Cardiovascular Physiology

LCPUFAs act as substrates for oxylipin production. LCPUFA metabolism to oxylipins occurs through three different pathways that are mediated by cyclooxygenase (COX), lipoxygenase (LOX), and cytochrome P450 enzymes (CYP) as a result of transcellular biosynthesis and coordination between distinct cell types [55,56,57]. Additionally, oxylipins can be produced by free radical catalyzed non-enzymatic lipid peroxidation [58,59]. High n-6 LCPUFAs and n-3 LCPUFAs intake is associated with higher levels of n-6 and n-3 LCPUFA-derived oxylipins, respectively [55]. In recent years, an active role for to maintain the tissues homeostasis has been attributed to the local generation of specialized pro-resolving mediators (SPMs). SPMs exert their effect by interacting with G-protein coupled receptor (GPCR), such as ALX/FPR2, GPR32, ChemR23, BLT1, and GPR18 [60,61], and modulate diverse biological responses [62] by regulating mitogen-activated protein kinase signaling, NF-κB pathways, AP-1 activation, and oxidative stress-related metabolism [63]. SPMs suppress the expression of adhesion molecules on leukocytes, endothelial cells, neutrophil chemotaxis, and IL-8 production. Frequently, SPMs are rapidly inactivated locally by eicosanoid oxidoreductases and prostaglandin dehydrogenase [64].

**Omega-6 mediators.** The most well-known oxylipins, the eicosanoids, are formed from ARA. ARA produces series-2 oxylipins via the COX pathway, resulting in the formation of prostaglandin G2 and subsequently to prostaglandin H2, which later gets converted to other prostaglandins and thromboxanes by specific prostaglandin and thromboxane synthases enzymes [65] (Figure 1). Prostaglandins make a huge oxylipins family with multiple biological functions such as inhibition of human platelet aggregation [66,67,68], neutrophil degranulation [69], innate immune response [70], activation of plasminogen activator inhibitor type-1 [71], and reducing pulmonary vascular resistance [72]. Prostaglandins can either have pro-inflammatory and vasoconstrictor effects [73] or anti-inflammatory and vasodilator effects [67,68], depending on to which receptor it binds [74]. Thromboxane A2 is an endothelium-derived oxylipin that potentially induces vasoconstriction and platelet aggregation [67] and thromboxane B2 is positively associated with high central blood pressure and multiple cardiovascular events [75,76].

In addition to the COX pathway, oxylipins are also formed by a second pathway involving LOXs that catalyze the formation of hydroxy fatty acid (5-, 12-, and 15-HETE are the most commonly described [77]) and their secondary metabolites, such as leukotrienes, lipoxins, resolvins, protectins, maresins, hepoxilins, and eoxins, via glutathione peroxidase [77]. ARA, via the LOX pathway, produces HpETEs, which can be further converted to leukotrienes (series-4), which are associated with atherosclerosis, endothelial dysfunction, and cytokine release [78]. Moreover, leukotrienes can also be converted to lipoxins [57], which play a role in the resolution of inflammation [79].

The third pathway that leads to oxylipins from LCPUFA metabolism involves a diverse array of membrane-bound cytochrome P450 (CYP) enzymes. ARA metabolism through CYP pathway activity results in the formation of HETE by CYP omega-hydroxylase activity. Cytochrome P450 can also act on ARA to synthesize epoxides, which could have vasodilator and vascular relaxing effects [80].

Limited information is available about LA-derived oxylipins. LA produces oxylipins through the LOX pathway (i.e., 13-HODE), COX pathways (i.e., 9-HODE), and epoxygenase activity of CYP (i.e., EpOME), as well as non-enzymatical pathways (i.e., 9-HODE; Figure 1), and the relative importance of these pathways needs to be explored. LA-derived oxylipins have been shown to attenuate platelet adhesion to endothelial cells [81], induce oxidative stress and a pro-inflammatory response in vascular endothelial cells [82,83], activate plasminogen activator inhibitor type-1 [71], and prevent platelets from adhering to human vascular endothelial cells [84,85]. However, the direct effect of LA-derived oxylipins on the heart has not been studied.

**Omega-3 mediators.** ALA, a precursor for n-3 LCPUFAs, can be metabolized to oxylipins by LOX [86], COX [55], or CYP cyclooxygenase activity [87] (Figure 2). ALA, metabolized via Cytochrome P450, results in epoxy fatty acids [87], and the levels of these epoxy fatty acids were decreased in adults with hyperlipidemia [88].

Similar to ARA, EPA metabolism through COX pathways generates oxylipins that includes series-3 prostaglandins (i.e., PGE3) and thromboxanes (i.e., thromboxane A3) [55]. However, relative to ARA, EPA is a weak substrate for COX enzyme [89]. Furthermore, EPA via LOX pathway results in leukotrienes (series-5) [90]. An in vitro study has shown that leukotrienes derived from both n-6 and n-3 LCPUFAs induce neutrophil lysosomal degranulation. However, n-3 LCPUFA-derived leukotriene (leukotriene B5) is less effective compared to n-6 LCPUFA-derived leukotriene (leukotriene B4) [91]. In addition, leukotriene B5 could be less inflammatory than leukotrienes 4-series [92]. The epoxides-fatty acids derived from EPA via CYP pathways [93], inhibits platelet aggregation, affect vasodilation, and have an antiarrhythmic effect in neonatal cardiomyocytes [94,95]. E-series resolvins synthesized by cytochrome P450 from EPA [64,96] have been shown to reduce neutrophil migration and inflammatory responses [97].

In addition to EPA, DHA is the other predominant omega-3 LCPUFA. DHA metabolism through LOX pathway leads to the production of maresins, resolvins (D-series), and protectins [55]. These endogenous lipid mediators including lipoxins, among other SPMs [95]. DHA-derived 4-hydroxy-docosahexaenoic acid (14-OH-22:6) potentially inhibits platelet aggregation [98]. Resolvins play an important role in inflammation, vascular biology, and platelet aggregation [99,100]. DHA-derived oxylipins can also be synthesized by COX [101,102] and cytochrome P450 epoxygenase. DHA derived epoxy-fatty acids [93] are known to decrease platelet aggregation and thromboxane A2 synthesis [95,103].

In general, oxylipins generated from n-3 LCPUFAs have lesser effectiveness compared to those derived from n-6 LCPUFAs. The effect of n-3 LCPUFA-derived oxylipins could be antagonized by oxylipins derived from n-6 LCPUFAs [104]. The n-3 LCPUFAs often compete with n-6 LCPUFAs for the same receptor and enzyme. The degree of membrane LCPUFAs incorporation might play an important role in determining the biological effect. Moreover, the enzyme activity could be regulated by increasing or decreasing the initial substrate within a given pathway.

## 3. The Role of SPMs in Cardiovascular Inflammation Resolution

The resolution of inflammation is tightly regulated by endogenously produced lipid mediators such as SPMs. These molecules can be lipoxins, resolvins, protectins, and maresins [105]. SPMs resolve the inflammation by alleviating the pro-inflammatory response, reducing neutrophil infiltration, and clearing the apoptotic cells through macrophages, and thus, enhancing the host defense. In addition, SPMs restrict the T cells actions, which is the main cellular responses involved in chronic inflammation [106,107]. SPMs are an important modulator of oxidative stress. There is evidence showing lipoxins inhibit leukocyte-dependent generation of reactive species [63]. LXA4 treated cardiomyocytes activate MAP-kinase and Nrf2 pathways [108], whose antioxidant properties are essential for cardiac protection [109]. In animal models, RvD1 reduces reactive oxygen species-mediated IL-1β secretion in macrophages [110] and protects against oxidative stress inflammation inhibiting neutrophil infiltration [111]. RvD1 and maresin 1 have been shown to regulate Nrf2-dependent expression of glutathione peroxidase and superoxide dismutase [112,113]. Kang et al. have reviewed a details summary about the role of EPA- and DHA-derived SPMs and Nrf2-antioxidative responses in cardiac fibrosis [109].

SPMs cause a shift in macrophages from M1 (pro-inflammatory) to M2 (anti-inflammatory) [114,115]. In addition to regulating the innate immune response, SPMs are also important in the adaptive immune response by reducing the NK cells cytotoxicity [116], decreasing memory B cell and antibody production [117]. D-series resolvins and maresin 1 dampen the cytokines production by activated CD8^+^ T cells, TH_1_, and TH_17_ cells and promotes the differentiation of CD4^+^ T cells into Treg cells, while inhibiting the generation of TH_1_ and TH_17_ from naïve CD4^+^ T cells [106,118]. Patients with chronic heart failure (CHF) have significantly reduced the plasma levels of RvD1 and pretreatment of mononuclear cells of patients with CHF with RvD1 or RvD2, which did not affect cytokine release from CD8^+^ and CD4^+^ T cells. This impaired T cells response was associated with reduced GPR32 expression [107]. The interaction between SPMs, their receptors, and their effect depend on the level of SPMs, cell type, and surrounding environment. Uncontrolled and unresolved inflammation can result in cardiovascular diseases, suggesting the critical role of SPMs. An extensive review regarding the lipid signaling pathways in cells and these effects on adult cardiovascular physiology and regeneration was covered by Wasserman et al. [119].

The etiology of cardiovascular disease involves a chronic inflammatory process driven by the formation of lipid-rich lesions in the vascular wall leading to myocardial infarction and stroke. Evidence about the role of SPMs in cardiovascular diseases came from human and animal studies. Serhan et al. have shown increased lipoxin levels in humans after angioplasty [120]. Overexpression of 15-LOX in macrophages resulted in a significant reduction of atherosclerosis in rabbits [121,122], while there was delayed atherogenesis in mice [123]. Administration of RvE1 attenuated atherogenesis in rabbits fed with a high fat and cholesterol diet, possibly by reducing the levels of C-reactive protein (CRP) [124].

The formation of a vulnerable plaque region (a subset of atherosclerotic plaques), with increased inflammation, oxidative stress, and necrotic areas as a result of increased cell death, might lead to acute atherothrombotic clinical events, such as myocardial infarction and stroke, probably due to a defective inflammation-resolution process [125,126,127]. Advance plaque regions in Ldlr−/− or ApoE−/− mice fed a high-fat and high-cholesterol diet displayed an imbalanced ratio of SPMs and pro-inflammatory lipid mediator ratio compared to early plaque regions [128,129]. Furthermore, SPM administration, including RvD2 and Maresin 1, or aspirin-triggered lipoxin A4, caused delayed atherosclerosis and resulted in a more stable-like plaque phenotype [129,130], probably due to decreased necrosis, oxidative stress, and increased fibrous cap thickness and helping in tissue repair processes.

Decreased blood flow can lead to tissue injury, as a result of the inflammatory response, due to leukocyte infiltration and ROS production [131]. This suggests that controlled leukocyte infiltration, as well as their removal from the site of injury, is important to protect the heart and myocardium during ischemia. Indeed, in mice, exogenous administration of RvD1 has been shown to improve cardiac function by reducing leukocyte infiltration and fibrosis [132]. Moreover, in a rat model of myocardial ischemia/reperfusion injury, RvE1 protects the rat heart by decreasing infarct size [133]. In addition to the heart, the protective effects of SPMs have been reported in other tissues, including the kidney and lung [134,135], possibly by reducing leukocyte recruitment.

Atherothrombosis is one of the clinical manifestations which leads to myocardial infarction and stroke [136]. Platelets and neutrophils aggregation are important for plaque inflammation [137]. Increased levels of circulating leukocyte-platelets aggregates in cardiovascular diseases suggest their contribution to pathogenesis. Moreover, elevated levels of platelet–monocyte aggregates have been used as an early marker of acute myocardial infarction [138]. It is possible that platelets and neutrophils/macrophage aggregates cause plaque formation and cardiovascular disease and are unable to produce enough SPMs. The levels of lipid mediators, specifically thromboxane and prostacyclin, are critical for platelet activation and thrombosis. In a randomized human-clinical trial, healthy subjects receiving aspirin at a recommended dose for patients with cardiovascular diseases (low dose = 81 mg) resulted in aspirin-induced 15-epi-LXA4, and the levels were negatively correlated with plasma thromboxane B2 levels, reducing the platelets activation [139]. Moreover, EPA-derived RvE1 has been shown to have an antiplatelet aggregation activity [140]. In response to acute inflammation, the self-limiting events such as platelets and neutrophils aggregation are critical and involve the biosynthesis of SPM to timely resolve the inflammation by reducing platelets and neutrophils aggregation, decrease cytokines production, and increase apoptosis [141,142].

SPMs, such as marsein 1, induce a pro-resolving platelet phenotype by increasing platelet aggregation and decreasing the levels of pro-inflammatory and pro-thrombotic mediators. All these data suggest the key role of SPM in promoting resolution inflammation as well as thrombosis during inflammation, and thus, could have potential therapeutic implications in cardiovascular diseases.

### SPMs in Infant Cardiovascular Health and Disease

LCPUFAs and oxylipins can modulate the balance for infant cardiac health and disease, by regulating the inflammation pathways. Gestational age and birth weight are important risk factors for the development of cardiovascular diseases [3,143,144], possibly by reducing endothelial function [145,146]. Prostaglandin E2 treatment is widely used in aortic coarctation [147], which is the third most common congenital cardiac lesion in preterm infants [148], by widening of the constricted ductal tissue within the aorta. Conversely, prostaglandin E1 in near-term infants resulted in a worsening of the aortic constriction [149]. Differences in pharmacokinetics and pharmacodynamics efficacy of prostaglandin therapy depend on the clinical course, the effects of surfactant-deficient lung disease, concurrent infections, myocardial insufficiency, and hemodynamic instability.

The ductus arteriosus is the fetal artery that connects the pulmonary artery and the aorta. The closure of the ductus arteriosus mostly occurs within three days of life in healthy term newborns as a result of increased oxygen level and reduced prostaglandin E2 [150]. However, in preterm infants, the ductus fails to close, resulting in patent ductus arteriosus (PDA). PDA is a heart problem that is common in preterm infants and causes morbidities such as bronchopulmonary dysplasia, intraventricular hemorrhage, and necrotizing enterocolitis and mortality [151,152]. The current strategy to treat ductus arteriosus involves the use of prostaglandins [151].

Inhibition of ARA-derived epoxides hydrolases showed beneficial cardiovascular effects, including vasodilation, anti-inflammation, anti-hypertrophy, and myocardial protection [153,154]. However, other data showed that epoxide hydrolase inhibition did not prevent cardiac remodeling or dysfunction [155], suggesting that targeting particular oxylipins seems to be insufficient for preventing cardiovascular events.

The E-series and D-series resolvins derived from n-3 fatty acids are important mediators in the resolution of inflammation [156], but their ability to modulate contraction of vascular smooth muscle is not known. In cardiovascular diseases, resolvins E1, D1, and D2 prevented constriction in human arteries induced by thromboxane [157] by resolving inflammation and preventing inappropriate vascular contractility.

Altered levels of resolvins are associated with cardiovascular disease onset, propagation, and systemic inflammation. Plasma levels of D-series resolvins were negatively correlated with decreased platelets and leukocyte activation [158]. Resolvin D1 administration in mice resulted in improved ventricular function following myocardial infarction by activating inflammatory response [132]. Most of the human studies regarding the effect of SPMs in cardiovascular disease have been done either in adult population or in children over 3 years. How these oxylipins act in neonatal cardiovascular remodeling and how nutritional intervention modulates this interaction need to be explored.

## 4. Relationship between SPMs and Infant Nutrition

Breast milk is an important source of LCPUFAs, fat-soluble vitamins, glycolipids, and lipoproteins [159]. Although their precise functions are not yet fully understood, it is known that the complex lipids in breast milk modulate infant physiology and signaling pathways, and thus, affect infant growth, development, and health [160]. Milk fat synthesis and its composition depend on maternal diet, body composition, substrates availability, and some hormones, including prolactin, growth hormone, and insulin.

In healthy women’s breast milk, throughout the first four weeks of breastfeeding, there is an increase in LA and ALA content, while oxylipins levels remain relatively stable throughout milk maturation [161,162]. Breast milk oxylipins are present at detectable levels (Table 1) [161], and are at higher concentrations than in plasma [163,164,165,166,167]. Higher levels of oxylipins in breast milk could be due to increased levels of the lipid mediators and their precursors in breast milk. Moreover, the presence of oxylipins in breast milk may reflect their translocation from maternal circulation across the mammary epithelium [168]. However, the production of oxylipins by leukocytes could be considered important, at least during the first weeks of the breastfeeding period, given the presence of leukocytes in breast milk [169]. The mechanism by which oxylipins would be higher concentrated in breast milk than blood is still unclear. Thus, there seems to be an enrichment of the lipid mediators and their precursors in breast milk. Oxylipins levels vary among breastfeeding mothers, and these differences could be the result of differences in maternal diet, gestational age, pre-pregnancy weight, and lactation stage, as well as the milk storage and testing methods used [170,171,172,173,174]. During the first month of lactation, breast milk contains higher levels of pro-inflammatory leukotriene B4 and anti-inflammatory and pro-resolving lipoxin A4, resolvin D1, and resolvin E1 [161,175]. Compared to whole blood, the levels of anti-inflammatory lipoxin A4 is two times higher in breast milk [161]. Due to the immunomodulatory functions, the lipid mediators, such as lipoxins, resolvins, and protectins, are thought to play an important role in cardiovascular diseases [90].

There are data that suggest a preferred accumulation of anti-inflammatory and pro-resolving oxylipins in breast milk, as seen by the bioactive products of ARA [161]. This fatty acid can be converted into pro- and anti-inflammatory lipid mediators, such as leukotriene B4 and lipoxin A4, respectively [175]. In breast milk, the average content of anti-inflammatory lipoxin A4 is twice as high as the pro-inflammatory leukotriene B4 [161]. Moreover, there are high concentrations of the precursor hydroxy fatty acids for the anti-inflammatory and pro-resolving lipid mediators present in breast milk. It is important to highlight that breast milk oxylipins play a key role in mobilizing and activating immune cells in newborns.

Oxylipin present at a concentration similar to lipoxin A4 in breast milk has been shown to inhibit intestinal and bronchial human epithelial cell inflammation by reducing neutrophil infiltration [176,177], suggesting that these levels should be enough to reduce inflammation in neonates. The pathophysiology of a given disease in infants and the prematurity might alter oxylipins in breast milk. Moreover, changes in breast milk storage temperature and duration of storage at a given temperature might affect the enzymatic activity [178,179], leading to increased lipid peroxidation. These changes might influence the quality and quantity of the bioactive components present in breast milk, including SPMs. Currently, administered parenteral lipid emulsion that is used as a common nutritional strategy in neonatal care mainly contains LA. However, some lipid emulsions also contain ARA, EPA, and DHA, which are crucial for infants to fight diseases. Studies have also shown the presence of n-6 derived oxylipins in lipid emulsions [180]. Lipids are susceptible to auto-oxidation, which depends on the composition of fatty acids, the level of unsaturation based on the number of double bonds present, delivery techniques, use of antioxidants, and environmental conditions [171,181]. Fatty acid oxidation could be through enzymatic pathways or non-enzymatic pathways. The non-enzymatic pathway is mediated by the presence of ROS and results in lipid peroxidation through sequential, self-propagating chain reactions [182,183]. During lipid peroxidation, the lipids are first converted to lipid hydroperoxides and then to aldehydes [184]. These products are highly reactive, and can quickly react with proteins, DNA, and phospholipids, causing deleterious effects [185]. However, the isoprostanoids, originating from ARA, neuroprostanes from DHA, and phytoprostanes from ALA are also molecules synthetized by the non-enzymatic pathway, which serve as markers of oxidative damage [186,187] but also exhibit bioactivities properties, such as antiarrhythmic protection [188] and mitochondrial homeostasis regulation in cardiac injury [189].

Lipid peroxidation appears to be directly related to the LCPUFAs content and inversely related to the α-tocopherol:LCPUFAs ratio in the lipid emulsion [190,191].

Although the lipid emulsion contains some antioxidants (ascorbyl-palmitate and α-tocopherol), they are possibly not enough to protect against lipid peroxidation [192]. An in vitro study has shown that L-carnitine possesses antioxidant and antiradical properties [193]. However, its effect on lipid peroxidation in parenterally fed infants has not been demonstrated. In addition, there are no guidelines for the maximum lipid peroxidation permitted in the emulsion [194]. Further studies will be necessary to assess the influence of storage conditions, shelf life after manufacturing, and appropriate antioxidant compounds on levels of LCPUFAs oxidation in parenteral emulsions.

Some studies have shown that consumption of a diet containing higher LA content increased the bioactive oxidized LA metabolites that have been implicated in chronic pain diseases [195]. The LA levels in infant lipid emulsions are usually high; whether this is detrimental for infant health still needs to be determined.

## 5. Research Gaps and Futures Perspectives

To date, there is a vast literature on acute, chronic, and non-resolving inflammation and their relationship with neonatal cardiovascular diseases. However, understanding of the molecular principles has not reached a level that permits accurate predictions [6]. The lipid mediators, including oxylipins, eicosanoids, and SPMs, are opening the way to a better understanding of the processes between health and disease. The nutritional role of lipids in neonatal cardiovascular health and their metabolic regulation makes it a therapeutic target in need of further research. Storage conditions, appropriate antioxidant levels, and LCPUFA oxidation in parenteral emulsions continue to be research questions. In addition, the critical window in which lipid nutritional intervention could be effective for cardiovascular development still needs to be determined.

However, it is not surprising that the same intervention may produce opposite effects at different times or in different neonatal populations. Basic biological investigations on lipids mediators and physiologically bioactive compounds would be necessary. Further studies are required to fully characterize the regulation of SPM receptor activity in cardiovascular cells, and the relationships between SPM synthesis, degradation, and downstream signaling events in vascular tissues, in both health and disease. Other questions that could complement the puzzle would be the contribution of epigenetics in modulating the SPM response.

## 6. Conclusions

Feeding high levels of unsaturated fatty acids increases the demand for antioxidants and enhances the possible membrane damage induced by pro-oxidants. The diet formulas should compensate for the addition of fatty acids with a corresponding increase in antioxidant contents. SPMs may interact at various levels in cardiovascular diseases. It is important to consider not only the immune response but also the structural cells of the vascular wall while considering the resolution of inflammation in cardiac diseases. SPMs are a new research strategy in cardiovascular disease and are poorly explored in newborns. These molecules can augment damage in response to injury, such as cell membrane rupture, platelet aggregation, increased cell adhesion, and vasoconstriction. However, SPMs are also involved in the vasodilatation process, anti-aggregation signaling, anti-inflammatory response, resolution of inflammation, and tissue repair and regeneration. The functions of SPMs are associated with the levels of their LCPUFA precursors, enzymatic activity, tissue physiology, and even organogenesis. It would be interesting to know the role that SPMs play in relation to metabolic variables in neonatal and adult cardiovascular physiology. This could open the door to new therapeutic and nutritional intervention strategies, both in adults and neonates. Future studies about how the resolution of inflammation is altered, and how SPM biosynthesis and receptor-mediated actions might be deregulated in neonatal cardiovascular diseases will help to develop novel pro-resolution therapeutics for cardiovascular diseases in neonates.

## Figures and Tables

**Figure 1 antioxidants-10-00933-f001:**
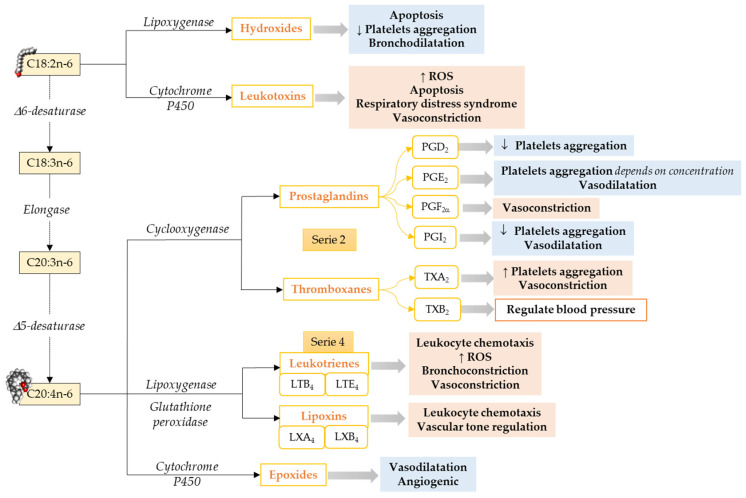
Pathways to synthesize n-6 LCPUFA-derived lipid mediators and their functions focusing on the cardiovascular system and vascular endothelial function. C18:2n-6, Linoleic acid; C18:3n-6, γ-Linolenic acid; C20:3n-6; Dihomo-γ-Linolenic acid; C20:4n-6, Arachidonic acid; ROS, Reactive Oxidative Species. ↓ = Decreases; ↑ = Increases. Modified from [55].

**Figure 2 antioxidants-10-00933-f002:**
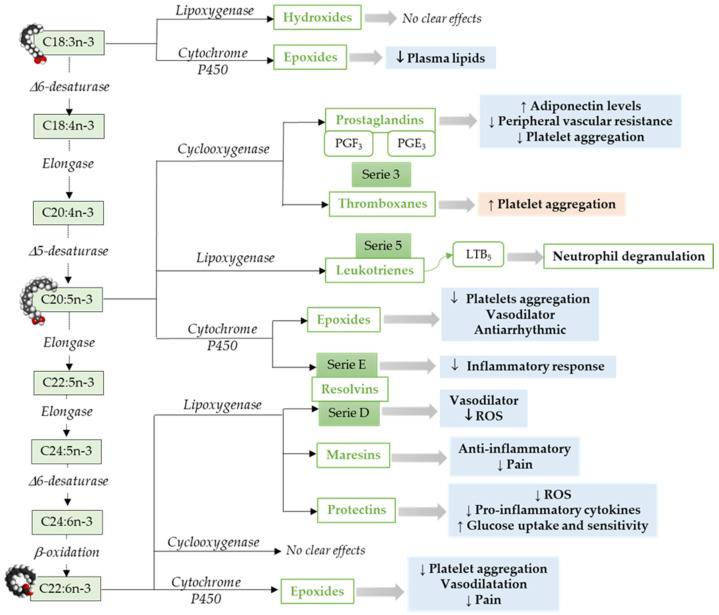
Pathways to synthesize n-3 LCPUFA-derived lipid mediators and its function focusing on the cardiovascular system and vascular endothelial function. C18:3n-3, α-Linolenic acid; C18:4n-3, Stearidonic acid; C20:4n-3, Eicosatetraenoic acid; C20:5n-3, Eicosapensaenoic acid; C22:5n-3, Docosapentaenoic acid; C24:5n-3, Tetracosapentaenoic acid; C24:6n-3, Tetracosahexaenoic acid; C22:5n-3, Docosahexaenoic acid; ROS, Reactive Oxidative Species. ↓ = Decreases; ↑ = Increases. Modified from [55].

**Table 1 antioxidants-10-00933-t001:** Levels of oxylipins found in breast milk.

Fatty Acid	Lipid Class	Oxylipin	Range (ng/mL)	Fatty Acid	Lipid Class	Oxylipin	Range (ng/mL)
LA	Hydroxides	13-HODE	50–70	ALA	Hydroxides	9-HOTrE	6.0–12
		9-OxoODE	150–200			13-HOTrE	2.0–6.0
	Leukotoxins	9,10-DiHOME	5.0–8.0	EPA	Epoxides	8,9-EpETE	0.0–1.0
		9,10-EpOME	25–75			14,15-EpETE	0.2–1.2
		12,13-EpOME	10–100			17,18-EpETE	0.8–1.3
ARA	Prostaglandins 2	PGF1a	0.3–1.2			14,15-DiHETE	0.4–0.8
		PGF2a	7.5–14.7			17,18-DiHETE	0.1–0.8
		PGE2	26.3–55.6			18-HETE	2.5–15.0
		PGD2	40–75.7		Resolvins E	RvE1	1.5–30
	Thromboxanes 2	TXB2	6.7–15.6			RvE2	19–45
	Leukotrienes 4	5-OxoETE	3.0–6.0			RvE3	26.5–62.6
		15-OxoETE	0.2–10	DHA	Resolvins D	RvD1	10–20
		5-HETE	15–45			RvD2	5.0–11
		8-HETE	1.0–3.2			RvD3	0.5–1.0
		11-HETE	1.0–3.5		Marsins	MAR1	1.0–3.0
		12-HETE	4.4–13		Protectins	PD1	0.2–0.7
		15-HETE	5.8–16.3		Epoxides	19,20-EpDPE	0.5–3.2
		LTB4	2.0–15.0				
	Epoxides	5,6-EET	10–30				
		8,9-EET	1.0–5.0				
		11,12-EET	1.5–2.5				
		14,15-EET	1.0–2.5				

Minimum and maximum detectable were extracted from [161,162,172,196].

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
