# Peer review of "Specialized Pro-Resolving Lipid Mediators in Neonatal Cardiovascular Physiology and Diseases"

_antioxidants, 2021, doi:10.3390/antiox10060933_

Round 1

Reviewer 1 Report

The present Review is well written and summarizes the current knowledge on SPMs and neontal cardiovasclular physiology and diseases.

I have the following comments that the authors need to address:

- The concept of resolution of inflammation and the fact that unresolved inflammation can lead to human diseases is not appropriately explained and it only briefly stated in this sentence "

- Uncontrolled inflammation and failure to re-41 solve the inflammation is the underpinning of several prevalent human diseases including cardiovascular diseases", without even any citations. It is a novel concept and should be nicely described and the authors should cite the following literature for this: PMID: 20303877; PMID: 29434586

- Since the entire Review is on SPMs, there should be a brief and dedicated paragraph on SPM metabolism, receptors and function. In the present review SPMs are only too quickly introduced at the end of paragraph 2.2 and in the subparagraph of omega-3 mediators and only 9 lines are on them. This is too short and should be more clearly described.

- The author stata that the role of SPMs is: "SPMs resolve the inflammation by alleviating the pro-inflammatory response, reducing neutrophil infiltration, clearing the apoptotic cells through macrophages and thus enhancing the host defense." Again, this is too short because SPMs do much more and the authors should describe into more detail all the bioactions of SPMs. These include shifting immune responses from pro-inflammatory to anti-inflammatory, promoting tissue repair and healing, limiting production of ROS (summarized in PMID: 31316721) also limiting the adaptive immunity responses of T cells, which are among the main cellular players involved in chronic inflammation, as reported in the following studies, including also chronic heart failure: PMID: 27559094; PMID: 30052486.

- Figure 1: The authors have to add Lipoxins (LXA4 and LXB4) from Arachidonic acid. These two lipoxins are also part of the family of SPMs and no mention of them in their origin/metabolism. Also PGE2 has to be added in the figure because is equally important as PGD2. PGF should be named as PGF2alpha (with the alpha in a single letter simbol in subscript next to the 2)

Author Response

The present Review is well written and summarizes the current knowledge on SPMs and neonatal cardiovascular physiology and diseases.

Response: Thank you for reviewing our manuscript and your suggestions. Below are our point-by point responses to your comments.

I have the following comments that the authors need to address:

  • The concept of resolution of inflammation and the fact that unresolved inflammation can lead to human diseases is not appropriately explained and it only briefly stated in this sentence "Uncontrolled inflammation and failure to resolve the inflammation is the underpinning of several prevalent human diseases including cardiovascular diseases", without even any citations. It is a novel concept and should be nicely described and the authors should cite the following literature for this: PMID: 20303877; PMID: 29434586.

Response: We agree with the reviewer’s comment. As suggested we added a paragraph and cited the two articles the reviewer mentioned (lines 41-49).

  • Since the entire Review is on SPMs, there should be a brief and dedicated paragraph on SPM metabolism, receptors and function. In the present review SPMs are only too quickly introduced at the end of paragraph 2.2 and in the subparagraph of omega-3 mediators and only 9 lines are on them. This is too short and should be more clearly described.

Response: Thank you for this astute observation and asking us to dedicate a paragraph on SPM metabolism, receptors and function. We agree with the reviewer’s comment and made the changes in the manuscript adding a new paragraph (lines 151-159) along with the new references.

  • The author stata that the role of SPMs is: "SPMs resolve the inflammation by alleviating the pro-inflammatory response, reducing neutrophil infiltration, clearing the apoptotic cells through macrophages and thus enhancing the host defense." Again, this is too short because SPMs do much more and the authors should describe into more detail all the bioactions of SPMs. These include shifting immune responses from pro-inflammatory to anti-inflammatory, promoting tissue repair and healing, limiting production of ROS (summarized in PMID: 31316721) also limiting the adaptive immunity responses of T cells, which are among the main cellular players involved in chronic inflammation, as reported in the following studies, including also chronic heart failure: PMID: 27559094; PMID: 30052486.

Response: We agree with the reviewer’s comment and made the changes in our manuscript by adding a paragraph (lines 297-319). We also added new references to support this information.

  • Figure 1: The authors have to add Lipoxins (LXA4 and LXB4) from Arachidonic acid. These two lipoxins are also part of the family of SPMs and no mention of them in their origin/metabolism. Also PGE2 has to be added in the figure because is equally important as PGD2. PGF should be named as PGF2alpha (with the alpha in a single letter simbol in subscript next to the 2)

Response: As suggested by the reviewer, we made these changes in Figure 1.

Reviewer 2 Report

Gila-Diaz and colleagues review highlights the role of SPMs in cardiovascular diseases. The group noted the different types of SPMs generated in vivo and connected them with cardiovascular physiology. The merit of this review is SPM in infant cardiovascular health and how nutrition i.e. breast milk have a role in infant’s supplementation of SPM precursors such as omega fatty acids. It is anticipated that there is a lack of research on SPM and infant for the group and more so on cardiac health to further comment in the review, making the weighting of the paper ‘light’ for this part. It is suggested that the authors add a small section on ‘research gap’ on SPM and neonatal cardiovascular health.

Comments:

  1. The paper is submitted to Antioxidant journal but does not explicitly describe antioxidant and neonates cardiovascular health. The review does not show if SPM have ‘antioxidant’ role in cardiovascular physiology and diseases. I would believe SPM have some control over Nrf2 translocation for ARE response (Antioxidants20209(12), 1259; https://doi.org/10.3390/antiox9121259).
  2. Line 137-138: how are they produced non-enzymatically? Free radical?
  3. Line 161-163: the authors are missing LOX-mediated arachidonic pathway which includes hydroxyeicosatetraenoic acid e.g. 12-HETE, 15-HETE
  4. Line 412-413: incorrect. Authors should note isoprostanes are generated too, especially neuroprostanes that have cardioprotective effect. PUFA non-enzymatic oxidation does not only release hydroperoxides and aldehydes, it’s a mix of isoprostanes as well. See:
  • Non-enzymatic oxidized metabolite of DHA, 4(RS)-4-F(4t)-neuroprostane protects the heart against reperfusion injury. Roy J, Fauconnier J, Oger C, Farah C, Angebault-Prouteau C, Thireau J, Bideaux P, Scheuermann V, Bultel-Poncé V, Demion M, Galano JM, Durand T, Lee JC, Le Guennec JY. Free Radic Biol Med. 2017 Jan;102:229-239.
  • Nonenzymatic lipid mediators, neuroprostanes, exert the antiarrhythmic properties of docosahexaenoic acid. Roy J, Oger C, Thireau J, Roussel J, Mercier-Touzet O, Faure D, Pinot E, Farah C, Taber DF, Cristol JP, Lee JC, Lacampagne A, Galano JM, Durand T, Le Guennec JY. Free Radic Biol Med. 2015 Sep;86:269-78
  • Enrichment of alpha-linolenic acid in rodent diet reduced oxidative stress and inflammation during myocardial infarction. Leung KS, Galano JM, Oger C, Durand T, Lee JC. Free Radic Biol Med. 2021 Jan;162:53-64
  • Non-invasive assessment of oxidative stress in preterm infants. Peña-Bautista C, Durand T, Vigor C, Oger C, Galano JM, Cháfer-Pericás C. Free Radic Biol Med. 2019 Oct;142:73-81.
  • Isoprostanes, neuroprostanes and phytoprostanes: An overview of 25years of research in chemistry and biology. Galano JM, Lee YY, Oger C, Vigor C, Vercauteren J, Durand T, Giera M, Lee JC. Prog Lipid Res. 2017 Oct;68:83-108

Author Response

Gila-Diaz and colleagues review highlights the role of SPMs in cardiovascular diseases. The group noted the different types of SPMs generated in vivo and connected them with cardiovascular physiology. The merit of this review is SPM in infant cardiovascular health and how nutrition i.e. breast milk have a role in infant’s supplementation of SPM precursors such as omega fatty acids. It is anticipated that there is a lack of research on SPM and infant for the group and more so on cardiac health to further comment in the review, making the weighting of the paper ‘light’ for this part. It is suggested that the authors add a small section on ‘research gap’ on SPM and neonatal cardiovascular health.

Response: Thank you for reviewing our manuscript and your suggestions. Below are our responses to your comments. In addition, research gaps and futures perspectives section was added in the new version of the manuscript.

Comments:

The paper is submitted to Antioxidant journal but does not explicitly describe antioxidant and neonates cardiovascular health. The review does not show if SPM have ‘antioxidant’ role in cardiovascular physiology and diseases. I would believe SPM have some control over Nrf2 translocation for ARE response (Antioxidants 2020, 9(12), 1259; https://doi.org/10.3390/antiox9121259).

Response: We have added about the role of SPMs as antioxidants, and its effect on Nfr2 signaling pathways (lines 301-302, 305-309) along with the references.

Line 137-138: how are they produced non-enzymatically? Free radical?

Response: Yes, oxylipins are generated from LCPUFAs through enzymatic reaction and free-radical-catalyzed non-enzymatic lipid peroxidation (PMID: 26805855). We have updated the information in line 148.

Line 161-163: the authors are missing LOX-mediated arachidonic pathway which includes hydroxyeicosatetraenoic acid e.g. 12-HETE, 15-HETE

Response: These oxylipins are added in the new version of the manuscript.

Line 412-413: incorrect. Authors should note isoprostanes are generated too, especially neuroprostanes that have cardioprotective effect. PUFA non-enzymatic oxidation does not only release hydroperoxides and aldehydes, it’s a mix of isoprostanes as well. See:

  • Non-enzymatic oxidized metabolite of DHA, 4(RS)-4-F(4t)-neuroprostane protects the heart against reperfusion injury. Roy J, Fauconnier J, Oger C, Farah C, Angebault-Prouteau C, Thireau J, Bideaux P, Scheuermann V, Bultel-Poncé V, Demion M, Galano JM, Durand T, Lee JC, Le Guennec JY. Free Radic Biol Med. 2017 Jan;102:229-239.
  • Nonenzymatic lipid mediators, neuroprostanes, exert the antiarrhythmic properties of docosahexaenoic acid. Roy J, Oger C, Thireau J, Roussel J, Mercier-Touzet O, Faure D, Pinot E, Farah C, Taber DF, Cristol JP, Lee JC, Lacampagne A, Galano JM, Durand T, Le Guennec JY. Free Radic Biol Med. 2015 Sep;86:269-78
  • Enrichment of alpha-linolenic acid in rodent diet reduced oxidative stress and inflammation during myocardial infarction. Leung KS, Galano JM, Oger C, Durand T, Lee JC. Free Radic Biol Med. 2021 Jan;162:53-64
  • Non-invasive assessment of oxidative stress in preterm infants. Peña-Bautista C, Durand T, Vigor C, Oger C, Galano JM, Cháfer-Pericás C. Free Radic Biol Med. 2019 Oct;142:73-81.
  • Isoprostanes, neuroprostanes and phytoprostanes: An overview of 25years of research in chemistry and biology. Galano JM, Lee YY, Oger C, Vigor C, Vercauteren J, Durand T, Giera M, Lee JC. Prog Lipid Res. 2017 Oct;68:83-108

Response: Thank you for this close review for clarity. We have included the changes as suggested (lines 467-471).

Reviewer 3 Report

This is a very clear and very well-written review concerning a very complex field. The authors succeeded in giving necessary background support to understand practical conclusions. This is for me an excellent review, very complete and up to date, on the one hand, very accessible to non-specialists interested in neonatal cardiology, on the other hand

I have only very minor suggestions:

  • lipoxin is mentioned a number of times in the text. It should be included in the two Figures. It was my reflex to go to the scheme and I guess that readers will have the same tendency. This should not be so complex and should not damage figures that have the merit to be particularly clear.
  • p.2, line 83: I do noth think that the assertion is correct, it is even misleading, suggesting that CL would provide fatty acids to mitochondria metabolism. In fact, 95 % is the percentage of LA present in CL, as indicated in ref 23. THis should be reformulated.
  • p.2, line 84: "should" or "must"?

Author Response

This is a very clear and very well-written review concerning a very complex field. The authors succeeded in giving necessary background support to understand practical conclusions. This is for me an excellent review, very complete and up to date, on the one hand, very accessible to non-specialists interested in neonatal cardiology, on the other hand.

Response: Thank you for reviewing our manuscript and your valuable suggestions. Below are our responses to your suggestions.

I have only very minor suggestions:

  • Lipoxin is mentioned a number of times in the text. It should be included in the two Figures. It was my reflex to go to the scheme and I guess that readers will have the same tendency. This should not be so complex and should not damage figures that have the merit to be particularly clear.

Response: The lipoxins are mainly derived from the omega-6 LCPUFAs precursors. As suggested we included lipoxins (LXA4 and LXB4) in Figure 1.

  • 2, line 83: I do not think that the assertion is correct, it is even misleading, suggesting that CL would provide fatty acids to mitochondria metabolism. In fact, 95% is the percentage of LA present in CL, as indicated in ref 23. This should be reformulated.

Response: We agree with the reviewer. We re-wrote the as “Cardiolipin (CL), a phospholipid that is exclusively present in the inner mitochondrial membrane, where it is essential for optimal functions of various key enzymes involved in mitochondrial respiration. Linoleic acid is one of the major fatty acids of CL in human myocardium. Evidence showed that CL must contain four linoleic  acid side chains for optimal mitochondrial function, and loss of linoleic acid content in CL resulted in human and rat model of heart failure.”

  • 2, line 84: "should" or "must"?

Response: Based on ref 23 it seems that it should be “must”.